# Exploring the mono-/bistability range of positively autoregulated signaling systems in the presence of competing transcription factor binding sites

**Rong Gao, Samantha E. Brokaw, Zeyue Li, Libby J. Helfant, Ti Wu, Muhammad Malik, Ann M. Stock**⊙*

Center for Advanced Biotechnology and Medicine, Department of Biochemistry and Molecular Biology, Rutgers University - Robert Wood Johnson Medical School, Piscataway, New Jersey, United States of America

* stock@cabm.rutgers.edu

**Data Availability Statement:** All relevant data, codes and parameter values used in this study are included within the manuscript, its Supporting

## Abstract

Binding of transcription factor (TF) proteins to regulatory DNA sites is key to accurate control of gene expression in response to environmental stimuli. Theoretical modeling of transcription regulation is often focused on a limited set of genes of interest, while binding of the TF to other genomic sites is seldom considered. The total number of TF binding sites (TFBSs) affects the availability of TF protein molecules and sequestration of a TF by TFBSs can promote bistability. For many signaling systems where a graded response is desirable for continuous control over the input range, biochemical parameters of the regulatory proteins need to be tuned to avoid bistability. Here we analyze the mono-/bistable parameter range for positively autoregulated two-component systems (TCSs) in the presence of different numbers of competing TFBSs. TCS signaling, one of the major bacterial signaling strategies, couples signal perception with output responses via protein phosphorylation. For bistability, competition for TF proteins by TFBSs lowers the requirement for high fold change of the autoregulated transcription but demands high phosphorylation activities of TCS proteins. We show that bistability can be avoided with a low phosphorylation capacity of TCSs, a high TF affinity for the autoregulated promoter or a low fold change in signaling protein levels upon induction. These may represent general design rules for TCSs to ensure uniform graded responses. Examining the mono-/bistability parameter range allows qualitative prediction of steady-state responses, which are experimentally validated in the *E. coli* CusRS system.

## Author summary

Cell survival in an ever-changing environment depends on appropriate responses to stimuli of different strengths. Bistability, i.e., two different stable states in responses to otherwise identical environments, can be beneficial in some systems but may need to be avoided by many signaling systems. Promoting or preventing bistable responses requires

information files and the following GitHub repository: http://github.com/winstongr/monostability.

**Funding:** A.M.S received funding from the National Institutes of Health (grant R35GM131727). The funders had no role in study design, data collection and analysis, decision to publish, or preparation of the manuscript.

**Competing interests:** The authors have declared that no competing interests exist.

specific architectures of gene regulatory networks as well as proper abundance and activities for the network building blocks. Here we use a mathematical model to study how the requirement for bistable or monostable responses places constraints on biochemical properties of autoregulated bacterial signaling systems. A particular focus is on how the relative abundance of transcription factor proteins to the number of their DNA target sites constrains the system design because competition for the limited TF protein molecules by DNA sites can promote bistability. We find that a strong binding affinity to the autoregulated TF promoter and low phosphorylation activities are preferred for monostability. We used an *E. coli* signaling system as an example and experimentally validated predictions of the model. Our results can help rationalize the regulatory features observed in naturally occurring systems as well as inform engineering of novel biological circuits for diverse signaling tasks.

## Introduction

Cells rely on accurate control of signaling systems to adapt to ever-changing environments. Appropriate responses to external stimuli often involve transcriptional regulation of gene expression, mediated by various interactions of transcription factors (TFs) with proteins, RNA and regulatory DNA sequences. A set of recurring network motifs, defined as "wiring" or connectivity patterns of TF interactions, can perform specific information-processing functions and constitute the basic building blocks of gene regulatory networks [1]. Architectures of these motifs, as well as the underlying biochemical properties that enable specific cellular functions, have been extensively studied [2–4]. These studies often consider only a limited set of transcription factor binding sites (TFBSs) that directly relate to a defined regulon although there is increasing evidence that many seemingly "non-functional" TFBSs exist for many TFs [5,6] and can compete with the autoregulated promoter to impact how individual motifs function [7–9]. Here we examine how the relative abundance of TFBSs affects the steady-state response and places constraints on the biochemical parameter range of positively autoregulated signaling systems.

Positive autoregulation is one of the most common motifs in gene regulatory networks. It has often been associated with bistability, i.e., existence of two stable steady states [1,3,10], which can facilitate cell differentiation and bet-hedging under challenging conditions [11]. Approximately 20% of TFs in *E. coli* activate their own expression according to RegulonDB [12]. More than 10 of these auto-activated *E. coli* TFs belong to the family of two-component systems (TCSs), one of the major signaling systems in prokaryotes [13,14]. A typical TCS involves a sensor histidine kinase (HK) that responds to environmental stimuli and regulates the phosphorylation level of its cognate response regulator (RR). HKs in prototypical TCSs are usually bifunctional with autokinase/phosphotransferase activities to increase RR phosphorylation and phosphatase activity to decrease phosphorylation (Fig 1A). Most RRs contain a DNA-binding domain and the phosphorylated RR is the active form that regulates transcription of response genes, many times including genes encoding the HK and RR themselves. Positive autoregulation is more prevalent than negative autoregulation in TCSs, yet bistable responses are not frequently seen while graded responses have been often observed [10,13,15]. The archetype TCSs with bifunctional HKs may have evolved to avoid bistability. It has been argued that bistability, with two cell populations having distinct responses to identical inputs, is often undesirable for many signaling systems [4]. A monostable, uniform and graded response may be favorable for many systems because it allows for a continuity of levels of gene expression or even activation of different sets of genes in response to different signal strengths.

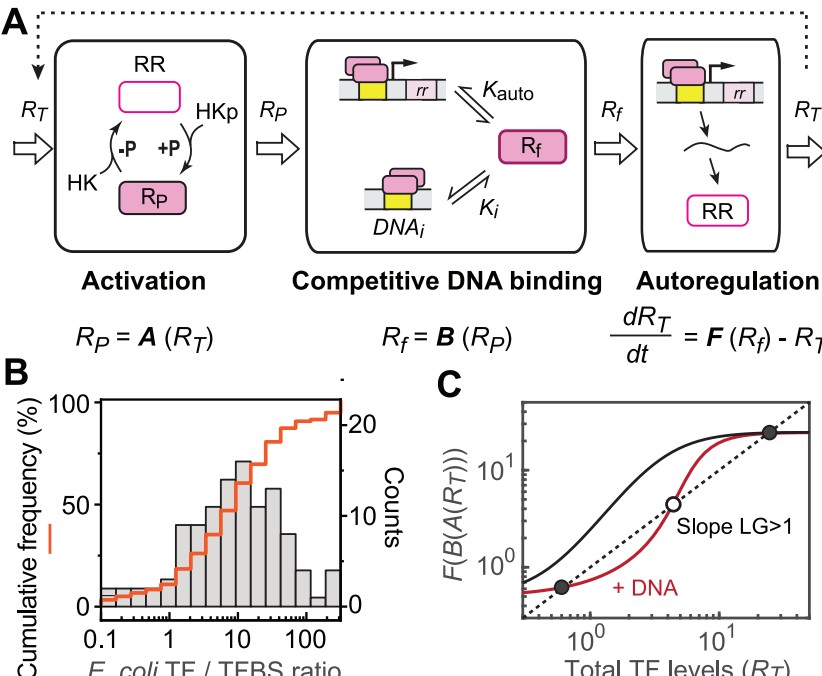

**Fig 1. Presence of competing TF binding sites can lead to bistability in positively autoregulated systems.** (A) Schematic of a typical autoregulated TCS. (B) Ratio of TF abundance to the number of TFBSs for *E. coli* TFs. The histogram shows the distribution of TFs with different TF/TFBS ratios and the red line indicates the cumulative frequency of TFs with ratios smaller than the corresponding histogram bin. (C) Steady states determined by the intersections of the autoregulated transcription function (solid lines) and the TF dilution/degradation (dotted line). An increase in competing TFBSs (+ DNA) can lead to multiple intersections, including two stable steady states (solid circles) and one unstable steady state (open circle), converting a monostable system (black) to bistable (red).

Achieving either the mono- or bistable response demands specific ranges of biochemical parameters, and the mono-/bistability ranges have been explored previously [4,16]. These studies focused on autoregulated pathways, while effects of TF binding to other genomic TFBSs were not considered. This is applicable for many TFs with TF protein abundance in great excess to the number of TFBSs. However, a considerable fraction of TFs has low TF/TFBS ratios with a large number of TFBSs and high binding demand. In *E. coli*, the median ratio of TF/TFBS is ~10 with many TFs having small ratios (Fig 1B). Nearly 20% of TFs have a TF/TFBS ratio smaller than 2, insufficient for protein molecules to occupy every DNA binding site, assuming a 2:1 binding stoichiometry that is used by most bacterial TFs. About one third of *E. coli* TFs have a TF/TFBS ratio smaller than 4, which is far from great excess. The curated TFBS data [12] are traditionally biased toward regulatory sites in proximity to promoters. With recent advances in global TF binding studies revealing many TFBSs in gene coding regions without apparent functions [5,6,17], the fraction of TFs with small TF/TFBS ratios will be even higher and a large fraction of TFs will be considered not to be in great excess to TFBSs [18]. Binding of TFs to these competing TFBSs can sequester the active TF molecules from the autoregulated TF promoter, and protein sequestration may cause ultrasensitivity and bistability [18–21]. Thus the number of TFBSs can impact the mono-/bistability range and further place constraints on biochemical parameters of the positively autoregulated signaling systems.

In this study, we focused on TCSs and incorporated a DNA binding competition model with autoregulation and RR phosphorylation/TF activation models [4,16,22] to examine the bistability-promoting effect from TFBS competition. TF autoregulation requires a high fold

change of transcription for bistability, however, TFBS competition can lower the fold change requirement. We show that bistability is sensitive to the relative binding affinities of competing TFBSs to the TFBS of the autoregulated promoter and the phosphorylation capacity of TCSs. Bistability can be largely avoided with a low fold change, a high TF affinity for the autoregulated promoter and low RR phosphorylation because they all weaken the overall positive feedback. Our model allows qualitative prediction of the mono-/bistability of TCS responses and such prediction was experimentally validated in the *E. coli* CusRS system.

## Results

### Model

The general model of a positively autoregulated signaling circuit is illustrated in Fig 1A. Based on biochemical functions and reaction timescales (see details in Methods and S1 Text), the system is decoupled into three modules: TF activation, competitive DNA binding and transcriptional autoregulation. The activation module defines TF activation that can vary for different signaling events such as binding of small molecules, protein-protein interaction or post-translational modification. Here we focus on phosphorylation of RR TFs in TCSs and the terms 'phosphorylation' and 'activation' are used interchangeably in this study. Output of the activation module, the concentration of active/phosphorylated TF molecules $R_P$, is a function of the total concentration of TF $R_T$. Active TF molecules are redistributed among different DNA binding sites via competitive DNA binding, affecting the concentration of free active TF molecules $R_f$. $R_f$ is the input of the autoregulation module, determining occupancy of the auto-activated promoter, thus the production rate of total TF levels $R_T$. As shown below, $A$, $B$ and $F$ represent individual functions of three modules:

$$R_P = A(R_T), R_f = B(R_P),$$

$$\frac{dR_T}{dt} = F'\left(R_f\right) - k_{\mathrm{dil}}R_T, \text{rewritten as:} \tag{1}$$

$$\frac{1}{k_{\mathrm{dil}}}\frac{dR_T}{dt} = F\left(R_f\right) - R_T = F(B(A(R_T))) - R_T,$$

where $k_{\mathrm{dil}}$ is the protein degradation or growth dilution rate.

Steady states correspond to intersections of the TF dilution curve and TF production function $F$ (Fig 1C). Bistability is associated with an ultrasensitive response function that has multiple intersection points. Existence of two stable steady states requires existence of an unstable steady state. As demonstrated previously [3,16,23], a necessary condition for bistability, or existence of an unstable steady state, is that the slope of the response function on the log-log dimension is larger than 1. Large values of the slope correspond to ultrasensitive or steep transitions in the response. Such slope has been termed open-loop logarithmic gain (LG) [16]. The overall gain $\mathrm{LG}_3$ is the product of LGs from three modules:

$$\mathrm{LG}_3 = \frac{d\mathrm{log}F}{d\mathrm{log}R_T} = \frac{R_T}{F}\frac{dF}{dR_T} = \left(\frac{R_f}{F}\frac{dF}{dR_f}\right)\left(\frac{R_P}{R_f}\frac{dB}{dR_P}\right)\left(\frac{R_T}{R_P}\frac{dA}{dR_T}\right) = \mathrm{LG}_F\mathrm{LG}_B\mathrm{LG}_A \tag{2}$$

where LGs of individual modules are defined as:

$$\mathrm{LG}_F = \frac{d\mathrm{log}F}{d\mathrm{log}R_f} = \frac{R_f}{F}\frac{dF}{dR_f}, \mathrm{LG}_B = \frac{R_P}{B}\frac{dB}{dR_P} = \frac{R_P}{R_f}\frac{dB}{dR_P}, \text{and } \mathrm{LG}_A = \frac{R_T}{A}\frac{dA}{dR_T} = \frac{R_T}{R_P}\frac{dA}{dR_T}.$$

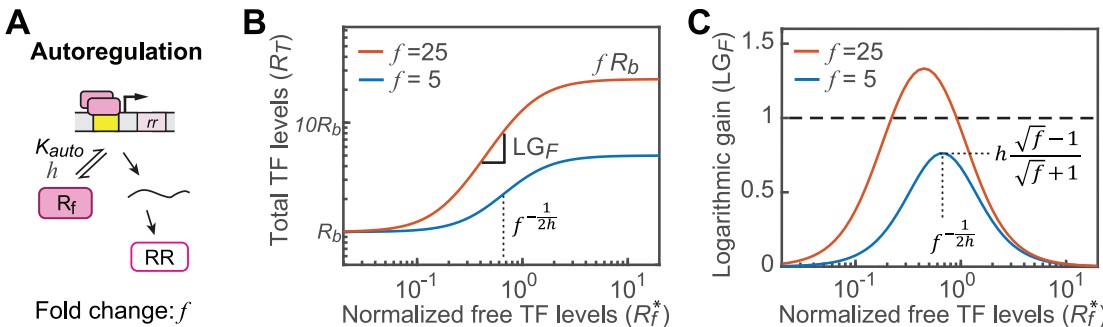

**Fig 2. Low values of the transcriptional fold change of the autoregulated promoter are advantageous to avoid bistability.** (A) The autoregulation module and the relevant parameters. The fold change (*f*) and the binding cooperativity (*h*) determine expression levels of the autoregulated TF (B) and the logarithmic gain LG_F (C). A high *f* value (orange) results in the maximal logarithmic gain exceeding 1 (dashed line) and the system potentially can be bistable.

For simplicity, three modules are considered decoupled so the LG of each module only depends on a limited set of independent parameters. Exploring the parameter space that enables bistability is thus reduced to examining LGs of individual modules that allow the overall gain to be larger than 1. An $\mathrm{LG}_3$ value larger than 1 is a necessary condition for bistability. If the maximum of $\mathrm{LG}_3$ is smaller than 1, the system will be monostable. Details of calculation of LG are included in Methods and S1 Text. Below we examine the autoregulation, DNA binding and phosphorylation modules in order, and discuss effects of individual parameter values on the monostability range.

## Monostability requires low transcriptional fold change in autoregulation

The module of transcription autoregulation (Fig 2A) has been extensively investigated [4,16] and transcriptional fold change of the autoregulated promoter greatly impacts the bistability range. Steady-state total TF levels $R_T$ range between the basal TF expression level $R_b$ and the fully induced level $fR_b$ (Fig 2B) where *f* is the fold change of transcription. Autoregulated transcription depends on promoter occupancy, which is determined by the concentration of free active TF $R_f$, binding affinity $K_{\mathrm{auto}}$ and binding cooperativity *h*:

$$R_T^* = R_b^* \frac{1 + f(R_f/K_{\mathrm{auto}})^h}{1 + (R_f/K_{\mathrm{auto}})^h} = R_b^* \frac{1 + fR_f^{*h}}{1 + R_f^{*h}} = F\left(R_f^*\right), \tag{3}$$

in which all concentration parameters are normalized to dimensionless values by dividing by $K_{\mathrm{auto}}$ to reduce complexity,

$$R_T^* = R_T/K_{\mathrm{auto}}, \quad R_f^* = R_f/K_{\mathrm{auto}}, \quad R_b^* = R_b/K_{\mathrm{auto}}.$$

As described previously [4,16], the logarithmic gain LG_F can be derived from the above equations and the maximum is:

$$\max(\mathrm{LG}_F) = h \frac{\sqrt{f} - 1}{\sqrt{f} + 1}, \text{ when } R_f^* = f^{-1/2h}. \tag{4}$$

It is apparent that high values of the fold change *f* or cooperativity *h* can result in high values of maximal LG_F that exceed 1 and promote bistability (Fig 2C). Considering that many RRs bind DNA as a dimer with the cooperativity $h = 2$, the bistability range is defined by the fold

change $f$. When $f$ is smaller than 9, the maximal $LG_F$ will be always smaller than 1, thus mono-stability favors low values of fold change. Bistability can potentially occur when $f$ is larger than 9. When $f \to \infty$, the maximal $LG_F$ is 2. For systems with high values of $f$, bistability can be prevented if high $LG_F$ values can be offset by coupled negative feedback, which reduces LG of the autoregulation module [24], or low LGs of the other two modules.

## Monostability range is affected by binding competition between the autoregulated promoter and other TFBSs

It has been shown previously that TFBS competition can lead to bistability in positively autoregulated circuits [18,19]. Here, we quantitatively evaluate how individual parameters of DNA binding competition impact the mono-/bistability range. To model the DNA binding module, we consider an extremely simplified scheme with all other DNA sites being identical with a cooperativity at 2 and a binding affinity $K$ (Fig 3A). All concentration and affinity parameters are normalized by $K_{auto}$ to ensure consistency with the autoregulation module.

The amount of competing TFBSs $D^*$ and the relative TF affinity $K^*$ determine the extent of competition that the autoregulated promoter faces (Fig 3 and S1 Fig). When there are no other TFBSs ($D^* = 0$), the model is similar to previous analyses that assume a great excess of TF molecules to the number of TFBSs. The concentration of free unbound phosphorylated TF ($R_f$) is approximately equal to the concentration of phosphorylated TF $R_P$, and the logarithmic gain

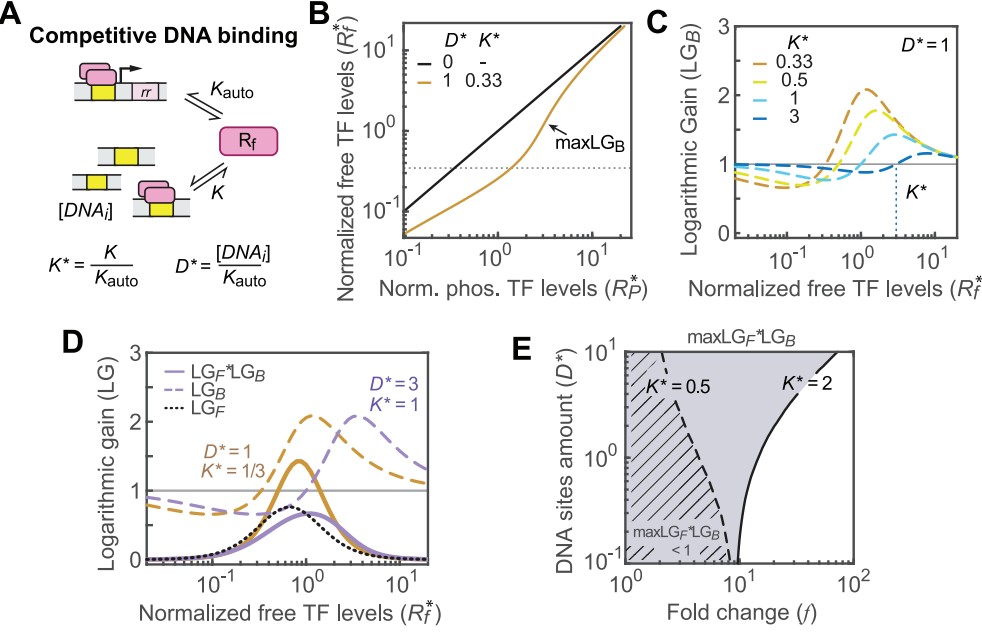

**Fig 3. The monostability range is greatly affected by DNA binding sites competition.** (A) A schematic of the DNA binding module. Binding competition is determined by the number of TFBSs and the relative binding affinity of TFBSs. (B) Dependence of $R_f^*$ on $R_p^*$ in the absence ($D^* = 0$) and presence ($D^* = 1$) of competing TFBSs. The horizontal dotted line indicates $R_f^* = K^* = 0.33$. (C) Effects of the relative TFBS affinity on the logarithmic gain $LG_B$. The horizontal gray line corresponds to $LG_B = 1$ when decoy TFBSs are absent. (D) Combined logarithmic gains of the autoregulation and DNA binding modules. Even with similar maximal $LG_B$ values (purple and brown dashed lines), the combined LG (solid lines) varies greatly depending on the relative TFBS affinity $K^*$. (E) A phase diagram showing the parameter space for monostable systems with the maximal combined LG <1. Solid and dashed lines represent the contour lines with $maxLG_FLG_B = 1$ at indicated $K^*$ values. When the relative affinity of TFBSs is weak ($K^* = 2$), a large monostable parameter space (gray shaded) is allowed, while stronger affinity ($K^* = 0.5$) leads to a smaller monostable space (striped).

$LG_B$ of the binding module is always 1 (Fig 3B and S1 Fig). When competing TFBSs are present, binding competition has opposite effects on $LG_B$ depending on the relative values of $R_f^*$ and $K^*$:

$$LG_B : \begin{cases} > 1, & R_f^* > K^* \\ < 1, & R_f^* < K^* \end{cases}. \tag{5}$$

The presence of other TFBSs reduces $LG_B$ below 1 when $R_f^*$ is less than $K^*$. This can be understood as these competing TFBSs serving as a "sink" to buffer against accumulation of free active TF molecules. When $R_f^*$ exceeds $K^*$, the "sink" is gradually getting filled with more than half of TFBSs bound, the accumulation rate of free active TF increases and $LG_B$ becomes larger than 1. Once a large fraction of TFBSs is bound (S1 Text), $R_f^*$ becomes ultrasensitive to $R_P^*$ and $LG_B$ reaches the maximum (Fig 3B and 3C and S1 Fig). The maximum of $LG_B$ increases with a larger amount of TFBSs $D^*$ (S1 Fig) or lower values of $K^*$, i.e., stronger relative affinity of other TFBSs (Fig 3C), and can reach well above 1.

However, a large maximum of $LG_B$ does not necessarily result in a large overall gain or bistability; $LG_B$ and $LG_F$ are not always synergistic. The combined LG, i.e., the product of LGs from the autoregulation and binding modules, $LG_F LG_B$, also depends on whether the maximums of the two modules are achieved at similar $R_f^*$ levels. Eq (4) indicates the peak $LG_F$ value occurring at $R_f^* = f^{-1/2h}$, and the parameter space having $LG_B > 1$ ($R_f^* > K^*$) needs to overlap with the peak $LG_F$ region to boost the combined LG. This requires $K^*$ values less than or near $f^{-1/2h}$. Because $f^{-1/2h}$ is smaller than 1 for positively autoregulated systems ($f > 1$), low $K^*$ values correspond to relatively strong affinities for competing TFBSs. As shown in Fig 3D, high amounts of competing TFBSs with intermediate TF affinity ($D^* = 3$, $K^* = 1$) yield a similar maximum of $LG_B$ as intermediate amounts of TFBSs with strong affinity ($D^* = 1$, $K^* = 1/3$), but the $LG_B$ peak of the former (purple dashed line) occurs at a higher $R_f^*$ level and is not well aligned with the $LG_F$ peak of the autoregulation module (black dotted line). Thus, the combined LG remains below 1 while the latter, with high affinity (brown solid line), shows an increase of LG above 1. For TFBSs with a relatively strong affinity, e.g., $K^* = 0.5$, DNA binding competition can promote bistability even if the fold change $f$ is smaller than 9 and a limited parameter space is monostable with the maximal LG smaller than 1 (striped in Fig 3E). With a relatively weak affinity ($K^* = 2$), the bistability range shrinks (Fig 3E). Even with a large number of competing TFBSs, the system can lessen the bistability-promoting effect of TF sequestration with a high $K^*$ value. This is also true with TFBSs having binding cooperativity different from 2, although the mono-/bistability range can vary greatly with different cooperativities (S1 Fig).

For auto-activated signaling systems, a high $K^*$ value may be favored to avoid bistability caused by TFBS binding competition and TF sequestration. Because $K^*$ is the relative binding affinity between competing TFBSs and the autoregulated promoter, a high $K^*$ value can be achieved if the affinity for the autoregulated promoter $K_{auto}$ is among the strongest of all TFBSs. We assessed the relative affinity of autoregulated promoters by inspecting information content of individual binding sites or binding peak intensities in chromatin immunoprecipitation (ChIP) experiments (Fig 4). Information content of individual binding sequences reflects similarity to the overall binding matrix [25] and high information contents often correlate with high affinities [26]. For each of the three selected RRs with well-curated TFBSs, binding site information content of the autoregulated promoter (magenta diamond) is among the top 15% of all TFBSs (Fig 4A). Similar patterns are also observed in ChIP binding intensities for multiple auto-activated *E. coli* TFs curated in the prokaryotic ChIP database, proChIPdb [27]. For individual TFs, binding peak intensities for positively autoregulated promoters are among

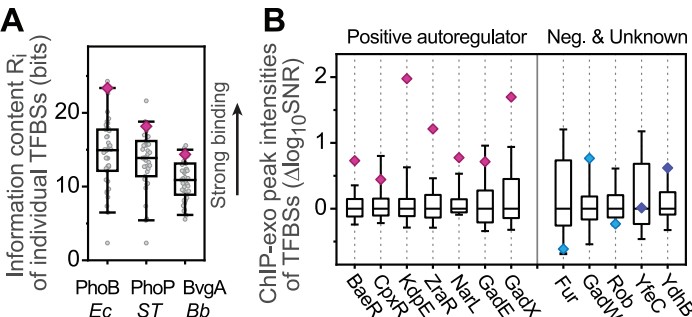

**Fig 4. Positively autoregulated TFs have strong affinities for their own promoters relative to their other binding sites.** (A) Distribution of individual information content ($R_i$) of TFBSs. $R_i$ of every TFBS (gray circles) was analyzed for PhoB from *E. coli*, PhoP from *Salmonella enterica* Ser. Typhimurium, and BvgA from *Bordetella bronchiseptica*. Values for TFBSs within the positively autoregulated promoters are shown as magenta diamonds. The numbers of TFBSs are: PhoB, 28; PhoP, 39; BvgA, 47. (B) Binding strengths of individual TFBSs reflected by ChIP-exo peak intensities. Logarithmic values of ChIP peak intensities (SNR) after subtracting the median ($\Delta\log_{10}$SNR) are shown as box plots. Binding peaks within the promoter of the operon encoding the TF are highlighted in color by their positive (magenta), negative (cyan) and unknown (blue) roles in autoregulation. The numbers of ChIP peaks for each TF are: BaeR, 178; CpxR, 113; KdpE, 78; ZraR, 107; NarL, 81; GadE, 35; GadX, 43; Fur, 66; GadW, 37; Rob, 534; YfeC, 50; YdhB, 29. Whiskers in all box plots indicate the range of 5–95%.

the highest of all binding regions across the chromosome (Fig 4B), suggesting that strong affinities for the auto-activated promoters may be selected for. The capability to lessen TFBS competition and avoid bistability with a strong affinity for the auto-activated promoter may be one of the reasons for such selection.

## Phosphorylation/dephosphorylation in a typical TCS limit the bistability range

To investigate how signal-regulated activation or phosphorylation of RRs impact the bistability range, we adopted a phosphorylation model described previously [22,28] based on a typical TCS that contains a bifunctional HK (Fig 5A). The steady-state phosphorylation level of an RR can be approximated with a simple function (see details in Methods and S1 Text) dependent only on the total RR concentration $R_T$, and composite parameters $Cp$ and $Ct$:

$$R_P^* \cong Cp^* \frac{R_T^* - R_P^*}{Ct^* + R_T^* - R_P^*}, \tag{6}$$

where all parameters with an asterisk are normalized by $K_{\text{auto}}$ to align with other modules. Signal-regulated activation of RRs is often modeled as an increase of $Cp^*$, leading to an increase of RR phosphorylation level that can be easily approximated with Eq (6). Such approximation relies on the assumption that the RR is in great excess to the HK. We assessed the effects of different RR/HK ratios on RR phosphorylation using the full scheme with typical TCS parameters (S2 Fig). Phosphorylation output with lower RR/HK ratios is not much different from that derived from Eq (6) except when the RR/HK ratio reaches 1. Based on proteomic studies in *E. coli* [29], a large fraction of TCSs have an RR/HK ratio much higher than 1 (S2 Fig), so the effects of RR/HK ratios on phosphorylation are expected to be small and are not considered here.

As noted previously [22,28], $Cp^*$ is the maximal phosphorylation level of the RR. Thus, we term $Cp^*$ as the phosphorylation capacity. The RR phosphorylation level $R_P^*$ increases with the total RR concentration $R_T^*$ and saturates at $Cp^*$ at high $R_T^*$ levels (Fig 5B). Correspondingly, the slope of $R_P^*$, $\text{LG}_A$, is high at low $R_T^*$ levels when $R_P^*$ is far from saturation, and

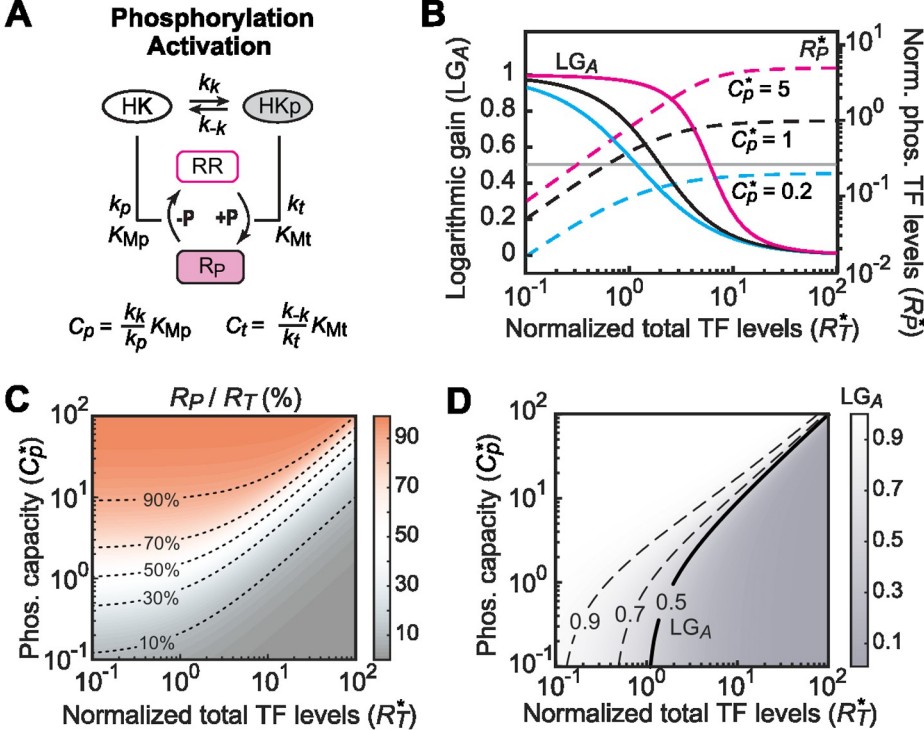

**Fig 5. Phosphorylation and dephosphorylation cycles in a typical TCS limit the bistability range.** (A) A schematic of the phosphorylation module. Autophosphorylation and auto-dephosphorylation of the HK are treated as pseudo first-order reversible reactions. The phosphotransferase and phosphatase activities of the bifunctional HK are modeled with Michaelis Menten kinetics. (B) Effects of the phosphorylation capacity $Cp^*$ on the logarithmic gain $LG_A$ (solid lines) and phosphorylation levels of RR $R_P^*$ (dashed lines). All curves are modeled with $Ct^* = 1$. The gray line indicates $LG_A = 0.5$; it intersects $LG_A$ curves at $Cp^*+Ct^*$. Higher $Cp^*$ values allow higher phosphorylation levels and larger ranges in which $LG_A$ remains high. (C and D) Dependence of the fraction of phosphorylated RR (C) and $LG_A$ (D) on $Cp^*$ and total TF level. Low $LG_A$ values (<0.5, gray) can lower the overall logarithmic gain, limiting the bistability of the autoregulation module.

gradually decreases to zero when approaching phosphorylation saturation. $LG_A$ can be derived:

$$LG_A = 1 - \frac{R_T^* - R_P^*}{\left(Ct^* + Cp^* - R_P^*\right) + R_T^* - R_P^*},$$

$$LG_A : \begin{cases} \leq 1 \\ < 0.5, \text{ when } R_T^* > Ct^* + Cp^* \end{cases} \tag{7}$$

The phosphorylation module alone does not promote bistability because $LG_A$ will never exceed 1. Instead, phosphorylation and dephosphorylation reactions can limit the bistability range if $LG_A$ is small. For example, in the absence of significant TFBS competition ($LG_B = 1$), the LG maximum of the autoregulation module is 2 for an autoregulated TF promoter with a cooperativity of 2 (see Eq (4)). The system can be monostable even with a large fold change $f$ if $LG_A$ is smaller than 0.5. The range of $LG_A$ smaller than 0.5 is defined by the sum of $Cp^*$ and $Ct^*$. Higher values of $Cp^*$ result in wider ranges of high $LG_A$ values, and thus have a less limiting effect on bistability; vice versa, lower values of $Cp^*$ result in systems more likely to be monostable.

In general, high $Cp^*$ leads to high phosphorylation levels (Fig 5B and 5C) while $Ct^*$ is negatively correlated with phosphorylation (S3 Fig). It has been shown that stimuli modulate the autokinase rate $k_k$ or phosphatase rate $k_p$ or both, which would all be reflected in changes of the phosphorylation capacity $Cp^*$. A high stimulus could boost phosphorylation by increasing the phosphorylation capacity $Cp^*$. Effects of $Cp^*$ and $Ct^*$ on $LG_A$ are illustrated in Fig 5D and S3 Fig. Parameter ranges that result in a high percentage of RR phosphorylation (red shaded in Fig 5C) give high $LG_A$ values close to 1 (white in Fig 5D), while ranges with low $LG_A$ values usually have low phosphorylation percentages (gray shaded in Fig 5C and 5D). A high phosphorylation percentage indicates a small difference between $R_T^*$ and $R_P^*$, and thus gives high $LG_A$ values according to Eq (7). Biochemically, the phosphatase activity of the bifunctional HK has a negative effect on phosphorylation, causing the phosphorylation level not to rise similarly to $R_T^*$ and offsetting the positive feedback from transcription. A high percentage of phosphorylation usually suggests low phosphatase activity, thus, less negative impacts on phosphorylation and stronger positive feedback, which promotes bistability.

The full parameter space for mono-/bistability is assessed by multiplying LGs from three modules to examine the overall $LG_3$ (Fig 6). For bistability with $LG_3 > 1$, the peak LG of the autoregulation and binding modules must align with the region where $LG_A$ is relatively large

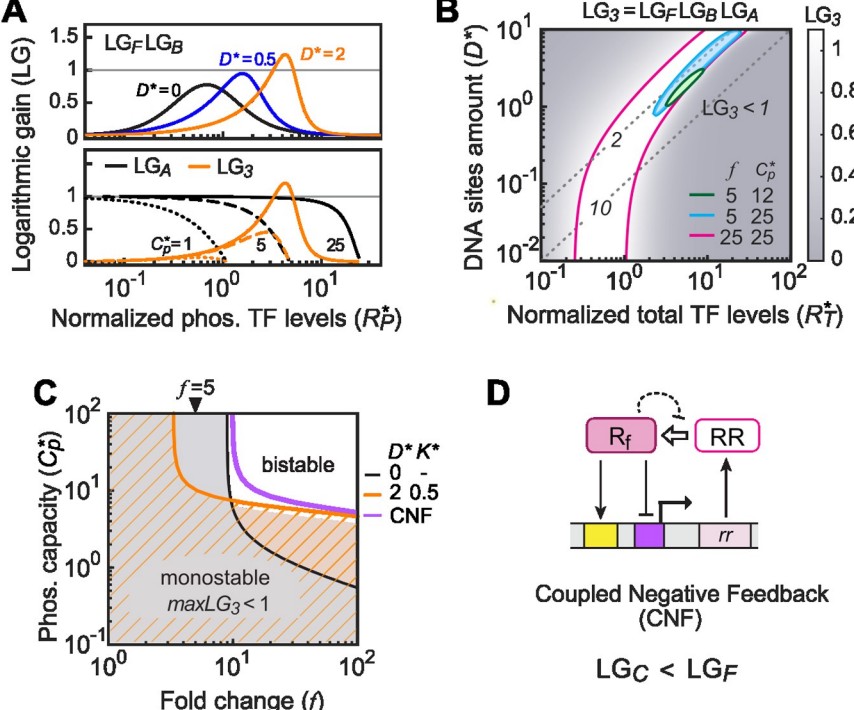

**Fig 6. The overall logarithmic gain $LG_3$ from the three modules determines the monostable parameter range.** (A) Overall gains $LG_3$ are limited by the phosphorylation capacity $Cp^*$. Increased numbers of TFBSs ($D^*$ values: black, 0; blue, 0.5; orange, 2) can elevate the maximal combined LG from the autoregulation ($f = 5$) and DNA binding modules ($K^* = 0.5$) above 1, but also shift the peak LG to higher phosphorylated TF levels (upper panel). This requires a high $Cp^*$ value to give a maximal overall $LG_3$ value above 1 (lower panel). (B) Effect of TFBS abundance $D^*$ on the overall $LG_3$. Regions bordered by colored lines indicate the parameter space with $LG_3 > 1$. Dashed lines with numbers indicate the TF/TFBS ratio. (C) The parameter space of $f$ and $Cp^*$ for monostability determined by max$LG_3$. Solid lines represent lines of max$LG_3 = 1$ under indicated conditions or feedback schemes. The gray shaded and orange striped areas indicate corresponding monostable ranges. For simplicity, the value of $Ct^*$ is arbitrarily set at 3 for all graphs shown here. (D) A schematic of the coupled negative feedback modeled in (C). The phosphorylated RR can bind to the activation site (yellow) and the repression site (purple) for coupled feedback regulation.

(Fig 6A). A large amount of competing TFBSs (orange lines in Fig 6A) promotes bistability, but requires a high level of phosphorylated TF molecules to occupy the DNA sites, thus a high phosphorylation capacity $Cp^*$. As shown in Fig 6B, a high $Cp^*$ together with a high fold change $f$ ($f = 25$) allows a large bistability range. For a system with a low value of fold change ($f = 5$), bistability can occur via TF sequestration in a limited parameter range (cyan). It demands a high TFBS amount $D^*$, a high TF level $R_T^*$ and a fairly small TF/TFBS ratio near 2 where the TF is not in great excess. A low phosphorylation capacity $Cp^*$ greatly reduces the bistability range (green). Fig 6C summarizes which parameter values of $f$ and $Cp^*$ lead to monostability with the maximum of LG$_3$ smaller than 1. In the absence of other TFBSs (gray shaded), high values of $f$ ($f>9$) enable bistability at high $Cp^*$ values while the system is monostable regardless of $f$ at low $Cp^*$ values. The presence of extra TFBSs changes the bistability range that allows for lower $f$ values but requires higher $Cp^*$.

Feedback control in TCSs can involve more complicated schemes with multiple coupled feedbacks [13]. Coupled negative feedback (CNF) has been shown to contribute to robustness, accelerate the response and shape the temporal dynamics [30–32]. Coupled negative feedback can also impact the mono-/bistability range. We consider a simple negative autoregulation scheme with the phosphorylated RR independently binding to a repression site within its own promoter (Fig 6D). The LG of the coupled regulation module, LG$_C$, is the sum of the LGs of positive and negative autoregulation. With the LG of negative autoregulation always negative (S4 Fig), the LG$_C$ is always smaller than the LG$_F$ of positive feedback alone. Thus, coupled negative autoregulation always decreases the overall LG and reduces the bistability range (purple line in Fig 6C and S4D Fig). Negative feedback can also occur through an RR activating production of proteins that inhibit RR phosphorylation (dashed line in Fig 6D), such as the *E. coli* RR PhoP, which activates expression of MgrB, repressing the HK PhoQ [31]. In such a scheme, coupled negative feedback functions to reduce $Cp^*$ at the steady state, which also limits the bistability range.

## Mono-/bistable response output in the presence of TFBS competition

The mono-bistability range shown in Fig 6C highlights the parameter space that can have a maximal LG$_3$ larger than 1 and allows bistability. However, a maximal LG$_3$ larger than 1 is only a necessary condition for bistability. Whether the maximal LG$_3$ can be achieved at steady states depends on other parameters, such as $R_b^*$. Steady states of an autoregulated system correspond to intersections of the RR production function $F$ and the RR dilution curve (gray line in Fig 7A). $R_b^*$ does not impact LG$_3$ but will determine the range of RR production levels and affect the steady-state intersections. A sample parameter set with $R_b^* = 2.5$ (red line) gives multiple intersections with one unstable and two stable steady states. In contrast, only one intersection or steady state exists for $R_b^* = 0.75$ (brown line) with the identical parameter set allowing LG$_3 > 1$. Deriving steady-state levels of $R_T^*$ and $R_f^*$ at different signal strengths, i.e., $C_p^*$, allows examination of the mono-/bistable responses.

Transcriptional responses depend on the steady-state levels of $R_f^*$ and are modeled as occupancy of RR-regulated promoters with the indicated affinity (Fig 7B and S5 Fig). In the presence of strong TFBS competition ($D^* = 4$, $K^* = 0.5$), the system displays a bistable response (red line in Fig 7B) at high $C_p^*$ values. Weakening the competition with $K^* = 2$ gives a monostable response (purple) to signals. In general, binding competition reduces the signaling capacity, i.e., "information capacity", of a signaling system [33]. A high $R_b^*$ value can ensure a sufficiently high $R_f^*$ that allows nearly full responses even with TFBS competition (S5 Fig). Similarly, a high fold change $f$ can lead to bistable responses (S5 Fig). $R_b^*$ also plays an important role in output because it impacts the concentration range of TFs. For some parameter sets

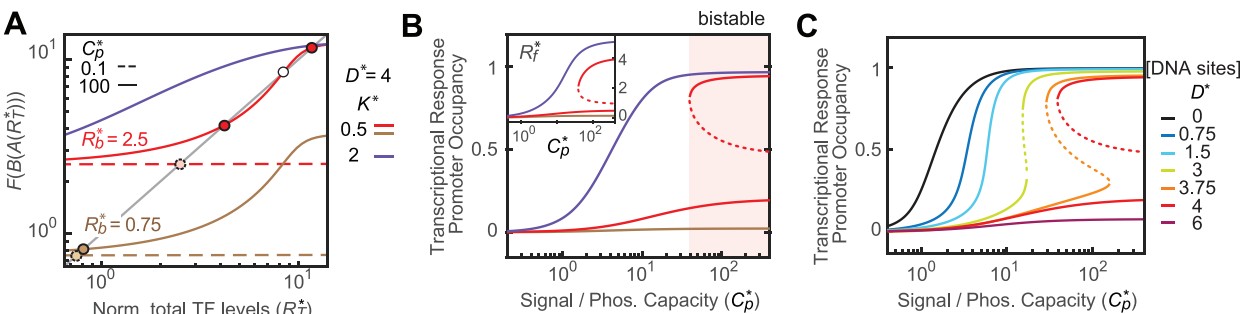

**Fig 7. Steady-state responses of the autoregulated system.** (A) Determination of steady states. Solid and dashed lines represent values of the RR production function $F(B(A(R_T^*)))$. Intersections with the RR dilution curve (gray diagonal line) indicate the stable (solid circles) or unstable (open circle) steady states. Varying $Cp^*$ values gives steady-state solutions to calculate signal-dependent responses. (B and C) Mono-/bistable responses affected by $R_b^*$, $K^*$ (B) and $D^*$ (C). Occupancy of the promoter with an affinity $K^*_{rep} = 1$ is used to represent transcriptional responses. The inset in (B) shows steady-state $R_f^*$ levels. Relevant parameters are: $D^* = 4$, $K^* = 0.5$, and $R_b^* = 2.5$. All curves shown in this figure are modeled with $f = 5$. Stable and unstable steady states are shown as solid and dotted lines, respectively.

that give a monostable steady state (brown line), a low $R_b^*$ coupled with a low fold change may result in insufficient $R_f^*$ and nearly no transcriptional response across different signals. Fig 7C illustrates how increasing amounts of TFBSs convert the monostable response of an autoregulated system with a low fold change ($f = 5$) to a bistable response at high amounts of TFBSs. This is experimentally explored in the *E. coli* CusSR system using increasing numbers of decoy binding sites.

## Experimental examination of the mono-/bistability in the CusSR system

Our model reveals how individual parameters shape the mono-/bistability range of TCSs. With accumulation of multi-omics data that allow estimation of some parameter values, it becomes possible to make qualitative predictions about the mono-/bistable response for specific systems. To evaluate the applicability of our model, we experimentally examined the output response of the *E. coli* CusRS system (Fig 8A). CusS is an HK that senses environmental copper and CusR is an RR transcription factor regulating several genes, including *cusCFBA*, that are responsible for exporting and detoxifying copper ions [34]. Environmental $Cu^{2+}$ concentrations alter the phosphorylation capacity and modulate outputs.

Cellular CusR levels range from ~170 to more than 500 molecules per cell in different measurements [29,35,36] while about a dozen CusR binding sites across the genome have been identified [37]. This suggests that CusR is likely in excess to the binding sites and TFBS competition may not play a significant role in the wild-type system ($LG_B \approx 1$). The fold change *f* has been observed to be small with only a modest increase in CusR protein level upon copper exposure [36]. We estimated the value of *f* to be ~5 based on the *PcusR-yfp* transcription reporter (S6 Fig). Considering that the autoregulated *cusR* promoter contains a single binding site [34], the binding cooperativity *h* will not likely exceed 2 and $LG_F$ will be smaller than 1 given *f* <9. The overall gain will be smaller than 1, thus the wild-type (WT) system will be monostable. The system can become bistable by altering the relative abundance of TFBSs to boost binding competition and increase $LG_B$. This can be achieved by introducing additional CusR binding sites. Exact prediction of DNA binding competition is difficult because of the complexity of CusR bindng modes and the lack of biochemical characterization (see details in S1 Text). However, a number of TFBSs at the same order of magnitude as the number of CusR molecules is likely required to promote bistability.

Output response from a chromosomal *PcusC* reporter expressing a bright fluorescent protein, mGreenLantern [38] was measured using flow cytometry (S7 Fig). As predicted, in the

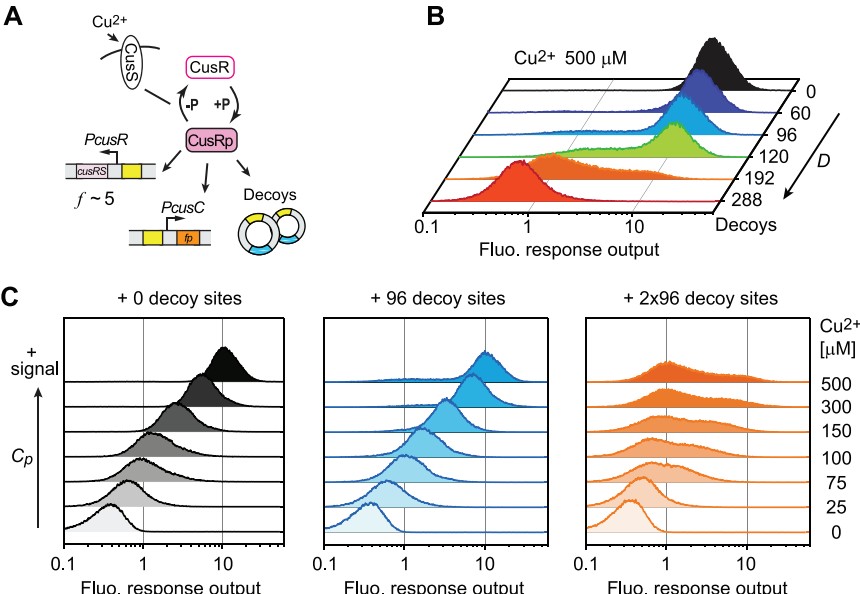

**Fig 8. Bistability in the CusRS system depends on the signal strength and the number of CusR DNA binding sites.**
(A) Experimental design for exploring bistability in the CusRS system. A chromosomal *PcusC-mGreenLantern*
reporter was used to track the response to environmental $Cu^{2+}$ concentration and plasmids carrying different number
of decoy CusR binding sites were used to alter the number of DNA binding sites. (B and C) Flow cytometry analyses of
YFP reporter output in response to different numbers of DNA decoys (B) and different $Cu^{2+}$ concentrations (C).

absence of extra TFBSs, the WT CusRS system displayed a monostable and graded response to
different $Cu^{2+}$ concentrations (black colored in Fig 7B and 7C). TFBS competition was exerted
by introducing plasmids carrying different numbers of decoy CusR binding sites. Bimodal
responses were observed with addition of 96, 120 and 192 decoy binding sites (Fig 7B and S7
Fig). Lowering $Cu^{2+}$ concentrations converted the bimodal output to monostable (central and
right panels in Fig 7C), e.g., a single uniform distribution of cell response was apparent at
intermediate $Cu^{2+}$ concentrations with 96 decoy sites. This is consistent with the prediction of
our model that lowering the phosphorylation capacity can reduce the overall LG and limit the
bistability range.

## Discussion

In this study, we investigated how the relative abundance of TFBSs to TF protein molecules
impacts the steady-state responses of positively autoregulated TCSs. The particular focus here
is on how TFBS competition constrains biochemical parameters in TCS phosphorylation and
autoregulation to ensure either mono- or bistable responses. With increasing numbers of TFs
discovered to be expressed in limited amounts relative to their binding sites, TF sequestration
by competing TFBSs appears to be common for many TFs and can have a wide range of effects
on response output, dynamics, noise and timing of gene expression [8,9,18,19,39–42]. Such
effects are tightly correlated with the abundance of TFBSs and TF proteins, which are subject
to considerable variations in cells, e.g., copy number variations in chromosome replication
and/or changes in protein levels under different growth conditions. Many signaling systems
need to operate under a variety of conditions with different relative abundances of TFBSs.
Here we characterized the mono-/bistability range and showed that a low fold change between
basal and maximal transcription rates of the autoregulated promoter, high TF affinity for the

autoregulated promoter and low phosphorylation can ensure monostability for positively autoregulated TCSs.

It has long been known that a large fold change ($f$ >9) is required for bistability [4,16]. TF sequestration by TFBSs enhances ultrasensitivity and positive feedback, lowering the fold change required for bistability. In many autoregulated TCSs, transcription of the TCS genes is under control of at least two promoters, a constitutive one for basal expression and an autoregulated one for maximal induction [13]. The fold change corresponds to the ratio of the strengths of the two promoters. Modulation of either promoter by mutation or stimuli can alter the fold change and the population distribution of the response. In the BvgA TCS, where a monostable and graded response is needed for precise control of the temporal order of gene expression [43], a promoter mutation that lowers the basal expression and increases the fold change has been observed to cause phenotypic bistability [44]. Assessment of bistability requires knowledge of the basal and maximal transcription levels to derive the fold change value. Even though the TF level under high stimulus does not always reach the maximal transcription level, the fold change value can be roughly approximated by the relative change of TF levels in response to high stimulus, which may be readily available in multi-omics datasets or easily measurable. Estimation of the fold change facilitates an initial assessment of the bistability, while the ultimate steady-state response will depend on the relative abundance of TFBSs and phosphorylation status. As demonstrated here for CusR, low fold change coupled with sufficient TF proteins for the TFBSs predicts monostable responses. Many TCSs appears to have high fold change values, e.g., 21 for PhoB [45], ~20 for LiaR [46] and ~50 for BvgA [47], yet graded rather than bistable responses are observed [43,48,49], likely because the phosphatase activity of the HKs limits the overall positive feedback.

Protein sequestration can cause ultrasensitivity and bistability. In TCSs, sequestration caused by formation of a non-functional "dead-end" complex between the HK and the RR, has been suggested to promote bistability and has been discussed in detail previously [10,50]. The "dead-end" complex provides a mechanism for an implicit feedback to enable bistability even in the absence of transcriptional feedback. The sequestration-based bistability studied here is dependent on the extent of competition between the binding site within the autoregulated promoter and other TFBSs.

Our analyses show that bistability can occur when the affinity of other TFBSs is strong relative to the affinity of the autoregulated promoter. In the presence of a large number of TFBSs, monostable responses can be achieved if these competing TFBSs have a relatively weak affinity in comparison to the autoregulated promoter. Therefore, a much stronger TF affinity for the autoregulated promoter relative to that of other TFBSs may be preferred to alleviate competition and promote monostability. Within available datasets that allow evaluation of the relative affinities, TF affinity for the autoreregulated promoter is always among the strongest of all TFBSs for positively autoregulated TFs, while such a trend is not observed in negatively autoregulated TFs. More global binding data are needed to evaluate this observation. Strong affinity for the autoregulated promoter is also advantageous for a fast transcriptional response, and weak affinity can result in an extremely slow response that does not reach the steady state within a physiologically relevant timeframe [4,9,26,30]. Benefits in both monostability and response speed may contribute to selection for strong affinity of TFs for binding sites within positively autoregulated promoters.

For sequestration-based bistability, high logarithmic gain or sensitivity occurs at a point where most TFBSs are bound by phosphorylated TFs, which, for most RRs, requires high phosphorylation levels of RR to occupy the TFBSs. The phosphatase activity of bifunctional HKs plays an essential role in setting the phosphorylation capacity. As noted previously [15,16], upregulation of the bifunctional HK via positive autoregulation also increases the

phosphatase activity, offsetting the positive feedback and limiting the bistability range. Any mechanism that lowers the phosphatase activity can potentially promote bistability in positively autoregulated TCSs. Bimodal and heterogeneous responses have been observed for a PhoQ phosphatase mutant [51,52] and PhoB cross-activated by a high amount of non-cognate HKs [53], both of which lack phosphatase activity. Phosphatase activities can protect TCSs from bistability and our analyses suggest that the "protection" range with low logarithmic gain is roughly correlated with low phosphorylation. Many autoregulated TCSs display a maximum percentage of RR phosphorylation around or below 50% [45,54–56]. It is likely that these autoregulated TCSs operate in a range with low RR phosphorylation to ensure monostability.

Lack of full RR phosphorylation may represent the cost for positively autoregulated TCSs to maintain a monostable and graded response. Similarly, other parameter values that facilitate bistability or monostability can also carry costs in different aspects of response. For example, a low fold change $f$ can reduce the propensity for bistability, but it restricts the range of RR protein levels, which can lead to a weak response to signals as shown in S5 Fig. High $K^*$ for other TFBSs limits the sequestration-based bistability but may also impact the binding of TFs to these TFBSs, among which are the regulatory targets for transcriptional responses. Satisfying different signaling needs often requires a concerted tuning of multiple parameters or even evolution of novel regulatory schemes [30,31]. Furthermore, TCS architectures can be much more complex than the canonical one analyzed here. Bistability and ultrasensitivity can originate from alternative configurations, such as phosphorelays and split HKs [10,57].

Overall, we investigate how TFBS competition impacts the mono-/bistability parameter space and how positively autoregulated TCSs can overcome bistability-promoting effects to ensure accurate responses. Our model is limited by a deterministic approach. The equilibrium method for modeling TF binding employed here will be inadequate for many TFs that are expressed in low copies and are subject to fluctuations in a noisy system. Stochastic simulation, such as the one described previously for decoy competition [42], may be required to further explore the constraints imposed by competing TFBSs. Furthermore, TFBSs can be promiscuous and may be bound by other TFs or DNA-binding proteins, such as nucleoid associated proteins. Binding competition will be altered when a considerable fraction of the TFBSs are not accessible due to occupancy by other proteins. We also assumed a universal slow degradation/dilution rate for all proteins and protein-DNA complexes. Differences in degradation rates between the two could impact predictions [58]. Nevertheless, our analyses provide a quantitative framework to account for the TFBS competition effect for designing synthetic signaling circuits or predicting the steady-state output for naturally occurring signaling systems.

## Materials and methods

### Model description

The model is based on a deterministic description of TCSs. Such an approach allows simple treatment of various reactions, especially the complex phosphorylation/dephosphorylation reactions of TCSs. The positively autoregulated signaling circuit is separated into three modules (Fig 1A). Reactions of different modules occur at different timescales with DNA binding being the fastest at a timescale of seconds, and transcriptional regulation the slowest, usually longer than 30 min and extending to hours. Phosphorylation and dephosphorylation of many TCSs occur at a timescale in min, between the other two modules [59,60]. Therefore, the three modules are considered as decoupled for simplicity. Further, growth dilution is ignored for the phosphorylation and DNA binding modules because their reaction timescales are much faster than typical bacterial growth rates. Details of decoupling approximation and mass action kinetics are described in S1 Text.

## Autoregulation module

Transcription and translation of the RR TF are lumped together to model the production of RR protein molecules, which depends on the binding probability of the promoter. Accounting for both production and growth dilution (see S1 Text for details), the ordinary differential equation (ODE) for total RR level $R_T$ is:

$$\frac{dR_T}{dt} = \alpha \frac{1 + f(R_f/K_{\text{auto}})^h}{1 + (R_f/K_{\text{auto}})^h} - k_{\text{dil}}R_T \tag{8}$$

in which $\alpha$ is the lumped protein production rate constant and $k_{\text{dil}}$ is the growth dilution rate. Normalizing all concentration parameters by $K_{\text{auto}}$ and letting $R_b^* = \alpha/(k_{\text{dil}}K_{\text{auto}})$,

$$\frac{1}{k_{\text{dil}}}\frac{dR_T^*}{dt} = R_b^* \frac{1 + fR_f^{*h}}{1 + R_f^{*h}} - R_T^* = F\left(R_f^*\right) - R_T^*. \tag{9}$$

The logarithmic open loop gain of the autoregulation module can be derived:

$$\text{LG}_F = \frac{R_f^*}{F\left(R_f^*\right)}\frac{dF\left(R_f^*\right)}{dR_f^*} = \frac{h(f-1)R_f^{*h}}{\left(1 + fR_f^{*h}\right)\left(1 + R_f^{*h}\right)}. \tag{10}$$

This is used to derive the LG values at different $R_f^*$ values and combined with LGs from the other two modules to explore the mono- and bistability range. The maximum value of $\text{LG}_F$ can be solved analytically:

$$\text{maxLG}_F = h\frac{\sqrt{f} - 1}{\sqrt{f} + 1}, \, when \, R_f^* = f^{-1/2h}.$$

In autoregulated TCSs, the RR and the HK are commonly transcribed from the same operon. Expression of HK protein molecules is modeled with a rate constant proportional to that of RR expression so that a constant ratio of RR to HK is maintained. Impact of RR/HK ratios on phosphorylation is discussed in the modeling of the phosphorylation module.

## DNA binding module

In the presence of competing TFBSs, TFs can be sequestered by DNA binding, lowering the concentration of free unbound active TF, $R_f$. Because many bacterial TFs bind DNA as dimers and each TFBS usually contains two half-sites, the binding cooperativity is set to 2 for the binding model discussed here. More analyses are provided in S1 Text for TFBSs with alternative binding cooperativity. An extreme case with all TFBSs having the same affinity $K$ is modeled. With the DNA concentration at $D$, $R_P$ is the sum of $R_f$ and DNA-bound phosphorylated TF molecules:

$$R_P = R_f + 2D\frac{R_f^2}{R_f^2 + K^2}. \tag{11}$$

All concentrations and relevant parameters are normalized by $K_{\text{auto}}$ to ensure consistency of variables from different modules:

$$R_P^* = \frac{R_P}{K_{\text{auto}}}, R_f^* = \frac{R_f}{K_{\text{auto}}}, D^* = \frac{D}{K_{\text{auto}}}, K^* = \frac{K}{K_{\text{auto}}}.$$

Both sides of Eq (11) are differentiated with respect to $R_P^*$ to obtain $dR_f^*/dR_P^*$. The LG of the DNA binding module can be calculated as:

$$\text{LG}_B = \frac{R_P^*}{R_f^*}\frac{dR_f^*}{dR_P^*} = 1 + \frac{2D^*R_f^*\left(R_f^* - K^*\right)\left(R_f^* + K^*\right)}{\left(R_f^{*2} + K^{*2}\right)^2 + 4D^*K^{*2}R_f^*} = 1 + \varphi\left(R_f^*\right)\left(R_f^* - K^*\right)$$

$$\text{where } \varphi\left(R_f^*\right) = \frac{2D^*R_f^*\left(R_f^* + K^*\right)}{\left(R_f^{*2} + K^{*2}\right)^2 + 4D^*K^{*2}R_f^*} > 0. \tag{12}$$

If $R_f^* > K^*$, we can conclude $\text{LG}_B > 1$. $\text{LG}_B$ values are computed with Eq (12) above and multiplied by $\text{LG}_F$ at individual $R_f^*$ levels. The maxima of combined LG values within a $R_f^*$ range are numerically computed to explore how the concentration and affinity of TFBSs affect the monostability range.

## Activation/phosphorylation module

Phosphorylation of the RR is modeled as described previously [18,22,28]. Degradation or growth dilution rates of TCS proteins are considered to be very slow in comparison to reaction rates within the phosphorylation module, and thus are ignored (see details in S1 Text). Concentration of the phosphorylated RR TF can be derived as follows:

$$R_P = Cp\frac{[\text{RR}]}{Ct + [\text{RR}]} \tag{13}$$

in which $Cp$ and $Ct$ are composite parameters defined in Fig 5A and [RR] is the concentration of the free unphosphorylated RR. From Eq (13), we can conclude $P_P < Cp$, and $Cp$ defines the maximal RR phosphorylation level. Based on mass conservation,

$$[\text{RR}] = R_T - R_P - [\text{HK} \cdot \text{RR}_\text{p}] - [\text{HK}_\text{p} \cdot \text{RR}]. \tag{14}$$

When the RR is in great excess to the HK, the last two terms for the HK-RR complexes can be ignored and $[\text{RR}] \cong R_T - R_P$. Substituting this into Eq (13) and normalizing all concentrations, we obtain:

$$R_P^* \cong Cp^*\frac{R_T^* - R_P^*}{Ct^* + R_T^* - R_P^*}, \left(Cp^* = \frac{Cp}{K_\text{auto}}, Ct^* = \frac{Ct}{K_\text{auto}}\right). \tag{15}$$

Solving Eq (15) gives:

$$R_P^* = \frac{1}{2}\left(Cp^* + Ct^* + R_T^*\right) - \frac{1}{2}\sqrt{\left(Cp^* + Ct^* + R_T^*\right)^2 - 4Cp^*R_T^*} = A\left(R_T^*\right). \tag{16}$$

This is the RR phosphorylation approximation equation described previously [22].

Logarithmic gain of the phosphorylation module can be obtained from Eq (16) (see S1 Text for details):

$$\text{LG}_A = \frac{R_T^*}{R_P^*}\frac{dR_P^*}{dR_T^*} = 1 - \frac{R_T^* - R_P^*}{\left(Ct^* + Cp^* - R_P^*\right) + R_T^* - R_P^*}. \tag{17}$$

Because $R_T^* \geq R_P^*, Cp^* \geq R_P^*$, the last term of Eq (17) is non-negative, and we can conclude:

$$\text{LG}_A \leq 1.$$

When $C_t = 0$,

$$R_P^* = \begin{cases} R_T^*, & R_T^* < Cp^* \\ Cp^*, & R_T^* > Cp^* \end{cases}, \mathrm{LG}_A = \begin{cases} 1, & R_T^* < Cp^* \\ 0, & R_T^* > Cp^* \end{cases}$$

It can also be derived from Eq (17) that:

$$\text{when } R_T^* > Cp^* + Ct^*, \mathrm{LG}_A < 0.5.$$

To calculate the overall LG, $R_P^*$ and $R_f^*$ are numerically computed for each $R_T^*$ level and corresponding LG values for three modules are multiplied to derive $\mathrm{LG}_3$.

## Parameter ranges

Parameter ranges are estimated based on experimental measurements from well-studied TCSs. The autoregulated fold change of RR levels has been documented for a few systems with values reaching 40 [45,47,61], a range of 1–100 is chosen for the fold change *f*. Most other parameters are normalized to $K_{\mathrm{auto}}$, which represents the DNA binding affinity of the RR, usually in the submicromolar range [30,62,63]. An affinity of 0.1 μM corresponds to ~60 molecules per cell (cell volume ~$10^{-15}$ L). RR abundance in *E. coli* ranges from 40 to 8000 molecules/cell [29], thus an $R_T^*$ range of 0.1–100 was selected. *Cp* and *Ct* values are based on previous measurements of the *E. coli* PhoBR system (*Cp* = 4 μM, *Ct* = 0.8 μM) [45].

## Bacterial strains and growth conditions

*E. coli* strain BW25113 is the parent strain for all strains constructed for TF activation assays. Similar to methods described previously [26], *PcusC* and *PcusR* promoters were amplified by PCR and cloned into pJZG146 that has a promoter-less *yfp* gene, resulting in pJZG157 and pJZG209 that were used for estimating fold change of transcription. To examine CusR activation in single cells, a *PcusC* promoter was fused with the mGreenLantern gene [38] to obtain stronger fluorescence than YFP. The fluorescent reporter was integrated into the HK022 phage attachment site in the chromosome of BW25113 using recombination strategies [64], yielding RU2118. Decoy CusR binding sites were made by annealing oligonucleotides (see details in S1 Text) with sequences based on previous analyses [34,65] and inserted into pRG475 [18], a plasmid with a copy number that has been determined previously and that can be further manipulated by arabinose. The resulting plasmids pCusRBS1 (1 site/plasmid), pCusRBS2 (2 sites), pCusRBS3 (3 sites) and pLH10 (0 site) [18] were introduced into RU2118 for flow cytometry analyses. All strains were grown at 37°C in LB broth or minimal A medium [66] supplemented with 0.4% (w/v) glucose and amino acids (40 μg/ml each).

## CusR activation assay

Bacterial cultures grown in minimal A medium overnight were used to inoculate fresh cultures with ~1:20 dilution. Fresh cultures in mid-log phase were harvested and resuspended in fresh media containing different concentrations of $CuSO_4$ (0–500 μM). The starting optical density measured at 600 nm ($OD_{600}$) was 0.15 and bacterial cultures were transferred to either 96-well plates for fluorescence reporter assays, or flasks for flow cytometry. For reporter assays measured by a Varioskan plate reader (Thermo Scientific), cells were incubated with shaking with fluorescence and $OD_{600}$ measured every 5 min. Data were processed as described previously [18]. For flow cytometry analyses, cells were incubated for 2 h and analyzed using a Gallios Flow Cytometer (Beckman Coulter). GreenLantern fluorescence (FL1) and CFP (FL10) were measured for ~60000 cells per sample. The binding site plasmids also carry a constitutively

expressed *cfp* gene as a control. Dead cells with no CFP expression were excluded from reporter analyses. Arabinose can alter the plasmid copy number and change CFP fluorescence. Thus, the relative plasmid copy number was calculated by comparing median CFP fluorescence from cells with 0.2% arabinose to cells without arabinose. The number of CusR binding sites was derived by multiplying the plasmid copy number by the number of binding sites on individual plasmids.

## Evaluation of TFBS binding strength

To assess binding affinities of TFBSs, individual information content ($R_i$) was calculated as described [25] for curated sets of TFBSs of PhoB from *E. coli* [26,67], PhoP from *Salmonella* [68], and BvgA from *Bordetella* [69]. Peak intensities from chromatin immunoprecipitation (ChIP) experiments can reflect binding strengths of a TF to various sites. ChIP peak intensities, defined as signal noise ratio (SNR), were obtained for autoregulated TFs from the prokaryotic ChIP database, proChIPdb [27]. After subtracting the median of logarithmic intensities of all identified peaks, the resulting values ($\Delta\log_{10}SNR$) were used to assess the relative binding strength of individual TFBSs. Definition of positive or negative autoregulation is based on RegulonDB [12]. Three other autoregulated TFs (ArgR, FNR and PuuR) in proChIPdb were not included due to the lack of peak intensity for the autoregulated promoter. Two other TFs with a fairly large number of TFBSs, YfeC and YdhB, were chosen because of the presence of binding sites within their own promoters [70] however, the regulatory role of these sites is unknown.

## Supporting information

**S1 Text. Supplementary methods.**
(DOCX)

**S1 Fig. Effects of TFBS competition on LG$_B$.** (A) Functional dependence of $R_f^*$ on $R_p^*$ with different amounts of TFBSs $D^*$. Arrows indicate where the maxima of LG$_B$ occur. (B and C) Increase of LGs by increased amounts of competing TFBSs. Dashed lines represent LGs of the binding module. Solid lines represent the combined LGs from the autoregulation and DNA binding modules, which are plotted on the scales of $R_f^*$ (B) and $R_p^*$ (C). (D) Differences in LG$_B$ with different binding cooperativity of competing TFBSs. Lowering TFBS cooperativity $l$ reduces the LG$_B$ maximum but expands the region with LG$_B$ >1. The formula indicates the $R_f^*$ value where LG$_B$ = 1. The dotted line shows the LG$_F$ from the autoregulation module as reference for aligning with LG$_B$. (E) A phase diagram showing the monostability range with different binding cooperativity $l$. The gray shaded area shows the monostability range (maxLG$_F$LG$_B$<1) with $K^*$ = 1 and $l$ = 2. The striped region shows the corresponding monostability range with $K^*$ = 0.5 and $l$ = 2. Colored lines represent the contour lines with maxLG$_F$LG$_B$ = 1 at different $l$ values and they indicate the borders of the mono-/bistability ranges.
(TIF)

**S2 Fig. Evaluation of the $R_p^*$ approximation model with different RR/HK ratios.** (A and B) Influences of RR/HK ratios on RR phosphorylation (A) and logarithmic gain LG$_A$ (B). Red lines illustrate the corresponding $R_p^*$ and LG$_A$ values obtained from Eqs (16) and (17) with our approximation model discussed in the main text, assuming the RR is in great excess to the HK. Colored lines represent simulated data with the indicated RR/HK ratios using the full model described in S1 Text. Significant deviation from the approximation model is only apparent with an RR/HK ratio of 1. Parameter values are as follows: $Cp^*$, 18.3; $Ct^*$, 2.1; $K_{auto}$,

100 molecules/cell ($10^{-15}$ L). (C and D) Protein molecule numbers of RRs and HKs in *E. coli*. Protein abundance data were obtained from ribosome profiling of the *E. coli* proteome [29] to derive the RR/HK ratios (D). Most of the RR/HK pairs have a ratio greater than 1.
(TIF)

**S3 Fig. Dependence of RR phosphorylation and LG$_A$ on *Ct*$^*$.** *Ct*$^*$ is negatively correlated with the phosphorylation level $R_p$$^*$ (A) and fractions of phosphorylated RR (B). Lower *Ct*$^*$ values lead to higher phosphorylation levels before saturation. (C and D) Effects of *Ct*$^*$ on the logarithmic gain LG$_A$. Dotted lines indicate the parameter values that give LG$_A$ = 0.5. Low LG$_A$ values (<0.5) can limit the LG of the autoregulation module regardless of the fold change *f*. All graphs are generated with *Cp*$^*$ = 1.
(TIF)

**S4 Fig. Coupled negative feedback decreases the overall LG.** (A) Illustration of a simple negative autoregulation scheme. A phosphorylated RR binds to the repression site within its own promoter with an affinity of $K_N$, leading to transcriptional repression with a fold change $f_N$. (B) The logarithmic gain of the negative autoregulation module. Because LG$_N$ is always negative, the coupled negative feedback (CNF) will lower the overall LG$_3$ (C) and shrink the bistability range (D) in comparison to the system with only a positive feedback (PF).
(TIF)

**S5 Fig. Signal-dependent transcriptional responses of the autoregulated system.** (A) Transcriptional response is defined as the promoter occupancy that is determined by $R_f$$^*$ and the binding affinity $K^*$$_{rep}$. $K^*$$_{rep}$ will be different for different promoters, such as the autoregulated RR promoter and the RR-regulated promoters that contain sites among the competing TFBSs. (B-D) Impacts of $R_b$$^*$ and $K^*$ on $R_f$$^*$, and subsequently the promoter occupancy. For systems with identical parameter sets (*f* = 5, *D*$^*$ = 4 and $K^*$ = 2), $R_b$$^*$ determines the concentration range of $R_f$$^*$ levels (blue or purple shaded areas in B) and the range of promoter occupancy in response to signals. In comparison to a weak TFBS affinity ($K^*$ = 2), a strong TFBS affinity ($K^*$ = 0.5) always leads to stronger TFBS competition, thus lower $R_f$$^*$ and lower promoter occupancy (C and D). For high $R_b$$^*$ and low $K^*$$_{rep}$ values, such differences can be small, and both (purple and red lines in C) can reach near full occupancy. (E) Impacts of $R_b$$^*$ and *f* on transcriptional responses. High *f* can result in bistable responses (pink) while low *f* gives monostable but extremely weak responses (navy).
(TIF)

**S6 Fig. Estimating the fold change of the autoregulated *cusR* promoter.** (A) Time course of YFP reporter output. BW25113 carrying pJZG209 (*PcusR-yfp*), pJZG157 (*PcusC-yfp*, positive control for Cu response) and pCL1920 (negative control with no *yfp*) were assayed for response to 0 or 500 μM CuSO$_4$. Fluorescence normalized by OD$_{600}$ was used as reporter output, with mean and standard deviations (std) from eight replicate wells shown. Results are from one representative of three independent experiments. (B) Estimation of the fold change. Steady-state output was computed as the average of output in the plateaued region (~60–120 min) of the time course data. Mean and std from three independent experiments are illustrated as bar graphs.
(TIF)

**S7 Fig. Flow cytometry analyses of fluorescence response output.** RU2118 (*PcusC-mGreen-Lantern*) carrying different decoy plasmids were assayed. All decoy plasmids carry a constitutively expressed CFP used for gating of the flow cytometry data (A) and estimation of plasmid copy number (B). A small population of cells with only background CFP fluorescence was

often observed, especially in the presence of high concentrations of toxic $CuSO_4$. None of these cells showed GreenLantern (GL) fluorescence. They are considered as dead cells or cells with impaired gene expression, thus were gated out with CFP histograms and excluded from further analyses. (B) Plasmid copy number estimation. The copy number of decoy plasmids has been determined as 96 in previous studies [18]. Addition of 0.2% arabinose (Ara) can inhibit replication of the plasmid origin and reduce the copy number, leading to reduced CFP fluorescence (left panel). The median of CFP fluorescence was used to derive the relative fluorescence. Solid and open circles represent CFP fluorescence of independent samples in the absence and presence of arabinose (middle panel). Arabinose reduced fluorescence to 62%, thus the copy number is estimated to be $0.62 \times 96 \approx 60$. Plasmid DNA extracted from corresponding cultures showed a similar value of the relative DNA amount (right panel), consistent with the relative copy number estimated from CFP. (C) Dot plots showing GL and CFP fluorescence of individual cells in response to 500 μM $CuSO_4$. Decoy numbers are computed based on the plasmids and conditions used as follows: 0, pLH10 (0 sites); 60, pCusRBS1 (1 site, +Ara); 96, pCusRBS1; 120, pCusRBS2 (2 sites, +Ara); 192, pCusRBS2; 288, pCusRBS3. (TIF)

## Author Contributions

**Conceptualization:** Rong Gao, Ann M. Stock.

**Data curation:** Rong Gao, Samantha E. Brokaw, Zeyue Li.

**Formal analysis:** Rong Gao.

**Funding acquisition:** Ann M. Stock.

**Investigation:** Rong Gao, Samantha E. Brokaw, Zeyue Li, Libby J. Helfant, Ti Wu, Muhammad Malik.

**Methodology:** Rong Gao.

**Supervision:** Ann M. Stock.

**Writing – original draft:** Rong Gao, Ann M. Stock.

**Writing – review & editing:** Ann M. Stock.

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
