## [Decision Letter · Decision Letter 0]

13 Sep 2022

Dear Dr. Stock,

Thank you very much for submitting your manuscript "Exploring the Mono-/Bistability Range of Positively Autoregulated Signaling Systems in the Presence of Competing Transcription Factor Binding Sites" for consideration at PLOS Computational Biology.

As with all papers reviewed by the journal, your manuscript was reviewed by members of the editorial board and by several independent reviewers. In light of the reviews (below this email), we would like to invite the resubmission of a significantly-revised version that takes into account the reviewers' comments. In addition, we woudl require you to deposit the documented codes ans dataseta in a citable (with DOI) repository such as datadryad, mendeley data, etc.

We cannot make any decision about publication until we have seen the revised manuscript and your response to the reviewers' comments. Your revised manuscript is also likely to be sent to reviewers for further evaluation.

Sincerely,

Oleg A Igoshin

Academic Editor

PLOS Computational Biology

Sushmita Roy

Section Editor

PLOS Computational Biology

Reviewer's Responses to Questions

**Comments to the Authors:**

Reviewer #1: Uploaded as an attachment

Reviewer #2: The author studied the autoregulatory two component systems combined with modular models and derivation of relationships between system’s mono-/bi-stability with transcription factor binding sites (TFBSs) competition, phosphorylation capacity, and fold changes. In particular the TFBSs competition and phosphorylation activity can shift the system from monostability towards space that is bistability prone. The models is well constructed, the derivation is quite elegant, and the experiments is straightforward and supporting the discoveries from the theory. Honestly, I quite enjoy reading the paper. I would recommend to publish this manuscript. There are some really minor points, they are not necessarily required to addressed in this manuscript (except the first one) but I would like to know how the authors would respond. I put them in the following:

1. Figure 1A, the middle box, it is a bit misleading with Pink box as Rf then the output of the middle module as Rf as well.

2. In general, many signaling or regulatory systems evolves towards bistability, ultrasensitive and bistable systems can usually serve as a noise filter to increase the fidelity of responding to fluctuating environments. Also, it should be much easier for a system to achieve monostability because instability requires both particular structures in corresponding parameter regime. But it seems that the authors think the systems are avoiding to become bistability, why uniform graded responses are needed from an evolutionary point of view?

3. Another interesting questions is that what if the other TFBSs are actually functional in other TCSs, (i.e. there are cross-talks between different TCSs) will that affect the rule discovered here for the overall responses?

Reviewer #3: This manuscript contains a (mostly theoretical) study of positively autoregulating two-component signaling systems. Earlier work has studied such systems extensively; this work, however, extends these analyses by considering competition between the transcription factor binding sites responsible for the autoregulation and other binding sites or "decoys". The presence of decoys can enhance the sensitivity of the feedback, and hence can result in bistable states under parameter regimes where the state is monostable in the absence of decoys. The goal of the manuscript is to determine the conditions for mono- and bistability in the presence of decoys. A limited experimental test of the theory is presented in which decoys are added to an existing system and its effect on the response is qualitatively compared to theory.

Technically, I think the study is well conducted and the manuscript is for the most part clearly written. Nevertheless I do have a few suggestions for improvement.

1. The authors have chosen an approach based on deterministic equations, even though it has been shown that stochasticity can affect bistability and bimodality and some previous models have used a stochastic approach to describe the effect of decoys (Burger et al). The authors should at least discuss the limitations of that approach. Also, I could not find one of the relevant references in this regard:

Anat Burger, Aleksandra M. Walczak, and Peter G. Wolynes

Influence of decoys on the noise and dynamics of gene expression

Phys. Rev. E **86**, 041920.

2. The function of two-component systems is (presumably) to allow cells to respond to changes in an environmental signal. Hence, I would say the "response" of the system is a function that describes the steady-state value of $R_T$ (or perhaps $R_P$) as a function of that signal. The authors seem to use the term "response" differently, because they sometimes state that the system has a mono- or bistable response where I would say they only showed that it is mono- / bistable for the given parameters and signal level. I recommend avoiding the term response in those cases.

3. Very much related to the previous point: throughout the manuscript I was sometimes confused because the analysis seems to (correctly) identify parameters/conditions that allow a monostable state, without verifying whether (a) the parameters would still allow the system to (significantly) respond to the environmental signal, and (b) bistability was avoided along the complete response curve (e.g., for all Cu2+ concentrations). For example, it is concluded that bistability can be avoided by choosing a small fold change $f$; but a small $f$ may also imply a weak response to changes in the signal. A similar point can perhaps be made about the parameter $Cp$. What I mean is: can the authors explain how the signal affects the system and how the parameters affect the over-all response to the signal? This may also help in connecting to the experiments, where the signal is the independent variable that has been manipulated.

## Minor comments

4. In Eq. (1) (and the corresponding equation in the supplement) the parameter $k_{dil}$ is eliminated after the second equality sign. Obviously, the left-hand and right-hand side of this equality sign are not equal; this is incorrect. (After the differential equation is equated to zero to find the equilibria, the $k_{dil}$ *can* be divided out, so this error has no consequences, but it should nevertheless be fixed.)

5. In line 145, I think "the necessary condition" should be "a necessary condition"; I'm sure we can state many other necessary conditions.

6. In lines 154-156 the reader gets the impression that the condition that LG3 > 1 is not only a necessary condition but also a sufficient one. This, I believe, is misleading: there are further requirements on $R_b^*$ , $K_{auto}$, and other parameters.

## Very minor comments

The comments below purely cosmetic (but easy to fix).

7 In several sentences, articles (the, a/an) seem to be missing.

8 In the equations, sometimes notation could be improved.

* The symbol * is not the mathematical symbol for multiplication.

* In equation (2) I see no reason to use partial derivatives.

* Variables consisting of multiple characters, such as $LG$ for linear gain, $HK$ for histidine kinase concentration, and so on, read as multiplications ($H \\cdot K$). In fact, the software used to typeset the equations interprets them as such and therefore inserts spaces between these symbols. It is in my opinion better replace them by a single symbol (possibly with a subscript label.

* Italic characters are reserved for variables or parameters, and hence standard functions such as $\\log(x)$ are normally set in Roman. Subscript labels should be Roman to distinguish them from indices (which are variables that can take on multiple values).

**Have the authors made all data and (if applicable) computational code underlying the findings in their manuscript fully available?**

Reviewer #1: Yes

Reviewer #2: **No: **The model and equations are provided, the mathematical derivations are clear. But there are no codes provides. It would be good to have some scripts for readers to reproduce the plots and simulations.

Reviewer #3: Yes

PLOS authors have the option to publish the peer review history of their article (what does this mean?). If published, this will include your full peer review and any attached files.

Reviewer #1: **Yes: **Satyajit Rao

Reviewer #2: No

Reviewer #3: No
---

## [Editor Report · Decision Letter 1]

14 Nov 2022

Dear Dr. Stock,

We are pleased to inform you that your manuscript 'Exploring the Mono-/Bistability Range of Positively Autoregulated Signaling Systems in the Presence of Competing Transcription Factor Binding Sites' has been provisionally accepted for publication in PLOS Computational Biology.

While it may bot be formally required by PLOS, we encourage the authors to release their data not only on Github but in a fixed form with DOI, e.g. with https://data.mendeley.com/ or similiar data repository. This will allow authors to create the version fo teh code that created the figures at the time of publication.

Best regards,

Oleg A Igoshin

Academic Editor

PLOS Computational Biology

Sushmita Roy

Section Editor

PLOS Computational Biology

---

## [Editor Report · Acceptance letter]

17 Nov 2022

PCOMPBIOL-D-22-00911R1 

Exploring the Mono-/Bistability Range of Positively Autoregulated Signaling Systems in the Presence of Competing Transcription Factor Binding Sites

Dear Dr Stock,

I am pleased to inform you that your manuscript has been formally accepted for publication in PLOS Computational Biology. Your manuscript is now with our production department and you will be notified of the publication date in due course.

With kind regards,

Zsofia Freund
